# Quantitative Assessment and Prognostic Associations of the Immune Landscape in Ovarian Clear Cell Carcinoma

**DOI:** 10.3390/cancers13153854

**Published:** 2021-07-30

**Authors:** Saira Khalique, Sarah Nash, David Mansfield, Julian Wampfler, Ayoma Attygale, Katherine Vroobel, Harriet Kemp, Richard Buus, Hannah Cottom, Ioannis Roxanis, Thomas Jones, Katharina von Loga, Dipa Begum, Naomi Guppy, Pradeep Ramagiri, Kerry Fenwick, Nik Matthews, Michael J. F. Hubank, Christopher J. Lord, Syed Haider, Alan Melcher, Susana Banerjee, Rachael Natrajan

**Affiliations:** 1Division of Brest Cancer, The Breast Cancer Now Toby Robins Research Centre, The Institute of Cancer Research, London SW3 6JB, UK; saira.khalique@icr.ac.uk (S.K.); sarah.nash@icr.ac.uk (S.N.); harriet.kemp@icr.ac.uk (H.K.); richard.buus@icr.ac.uk (R.B.); hannahcottom@googlemail.com (H.C.); Ioannis.Roxanis@icr.ac.uk (I.R.); naomi.guppy@icr.ac.uk (N.G.); Chris.lord@icr.ac.uk (C.J.L.); Syed.Haider@icr.ac.uk (S.H.); 2Division of Radiotherapy and Imaging, The Institute of Cancer Research, London SW3 6JB, UK; dave.mansfield@icr.ac.uk (D.M.); alan.melcher@icr.ac.uk (A.M.); 3Gynaecology Unit, The Royal Marsden NHS Foundation Trust, London SW3 6JJ, UK; julian.wampfler@rmh.nhs.uk (J.W.); Ayoma.Attygalle@rmh.nhs.uk (A.A.); Katherine.Vroobel@rmh.nhs.uk (K.V.); 4Department of Histopathology, The Royal Marsden NHS Foundation Trust, London SW3 6JJ, UK; 5Division of Molecular Pathology, The Institute of Cancer Research, London SM2 5NG, UK; Thomas.Jones@icr.ac.uk (T.J.); Michael.Hubank@icr.ac.uk (M.J.F.H.); 6Biomedical Research Centre, The Royal Marsden NHS Foundation Trust, London SM2 5PT, UK; katharina.vonloga@icr.ac.uk (K.v.L.); dipa.begum@icr.ac.uk (D.B.); 7Tumour Profiling Unit, The Institute of Cancer Research, London SW3 6JB, UK; pradeep.ramagiri@icr.ac.uk (P.R.); kerry.fenwick@icr.ac.uk (K.F.); n.matthews@imperial.ac.uk (N.M.); 8The CRUK Gene Function Laboratory, The Institute of Cancer Research, London SW3 6JB, UK; 9Division of Clinical Studies, The Institute of Cancer Research, London SM2 5NG, UK

**Keywords:** immune microenvironment, ARID1A, clear cell ovarian cancer, next generation sequencing, biomarker

## Abstract

**Simple Summary:**

Ovarian clear cell carcinoma (OCCC) is a rare subtype of epithelial ovarian cancer that has a poor response to chemotherapy. Here, we assessed the immunological features of a series of 33 OCCCs and identified an immune-related gene expression signature that correlated with a patient’s risk of recurrence. Additionally, using multiplex immunofluorescence, we assessed the spatial distribution and abundance of immune cell populations at the protein level and identified that tumour-associated macrophages (TAM) and regulatory T cells are excluded from the vicinity of tumour cells in low-risk patients, suggesting that high-risk patients have a more immunosuppressive microenvironment. We also found that TAMs and cytotoxic T cells were also excluded from the vicinity of tumour cells in *ARID1A* mutated OCCCs, suggesting that the exclusion of these immune effectors could determine the host response in *ARID1A* mutant OCCCs.

**Abstract:**

Ovarian clear cell carcinoma (OCCC) is a rare subtype of epithelial ovarian cancer characterised by a high frequency of loss-of-function *ARID1A* mutations and a poor response to chemotherapy. Despite their generally low mutational burden, an intratumoural T cell response has been reported in a subset of OCCC, with *ARID1A* purported to be a biomarker for the response to the immune checkpoint blockade independent of micro-satellite instability (MSI). However, assessment of the different immune cell types and spatial distribution specifically within OCCC patients has not been described to date. Here, we characterised the immune landscape of OCCC by profiling a cohort of 33 microsatellite stable OCCCs at the genomic, gene expression and histological level using targeted sequencing, gene expression profiling using the NanoString targeted immune panel, and multiplex immunofluorescence to assess the spatial distribution and abundance of immune cell populations at the protein level. Analysis of these tumours and subsequent independent validation identified an immune-related gene expression signature associated with risk of recurrence of OCCC. Whilst histological quantification of tumour-infiltrating lymphocytes (TIL, Salgado scoring) showed no association with the risk of recurrence or *ARID1A* mutational status, the characterisation of TILs via multiplexed immunofluorescence identified spatial differences in immunosuppressive cell populations in OCCC. Tumour-associated macrophages (TAM) and regulatory T cells were excluded from the vicinity of tumour cells in low-risk patients, suggesting that high-risk patients have a more immunosuppressive microenvironment. We also found that TAMs and cytotoxic T cells were also excluded from the vicinity of tumour cells in *ARID1A*-mutated OCCCs compared to *ARID1A* wild-type tumours, suggesting that the exclusion of these immune effectors could determine the host response of *ARID1A*-mutant OCCCs to therapy. Overall, our study has provided new insights into the immune landscape and prognostic associations in OCCC and suggest that tailored immunotherapeutic approaches may be warranted for different subgroups of OCCC patients.

## 1. Introduction

Ovarian clear cell carcinoma (OCCC) is a rare aggressive subtype of epithelial ovarian carcinoma (EOC), characterised by a distinct repertoire of clinical, histological and molecular features [1,2,3]. With the exception of early-stage disease, OCCC is associated with the poorest stage-adjusted prognosis when compared to other EOC subtypes and shows relative resistance to chemotherapy [4,5]. Thus, there is a clear unmet clinical need to identify additional treatment for those OCCC patients that show poor responses to chemotherapy, and identification of biomarkers for OCCC patient stratification.

OCCCs harbor high frequencies of *ARID1A* (AT rich interactive domain 1A) loss of function mutations [1,3], which lead to an aberrant cell cycle and loss of proliferation control [6,7,8], and are associated with endometriosis [3]. A number of studies have highlighted potential synthetic-lethal treatment strategies targeting tumours with loss of ARID1A [9,10,11,12,13,14,15,16,17,18,19]. There is emerging evidence that OCCC may respond to an immune checkpoint blockade. In the KEYNOTE−100 phase II clinical trial, patients with advanced recurrent OCCC had a 15.8% overall response rate to pembrolizumab, compared to 8.5% in unselected recurrent ovarian cancers [20]. Similarly, a smaller phase II clinical trial assessing the combination of nivolumab and ipilimumab induction followed by nivolumab maintenance showed a higher response rate and longer progression-free survival (PFS) when compared with nivolumab alone in platinum-resistant OCCC patients [21]. 

Mis-match deficient (MMR) microsatellite instability (MSI)-high OCCCs have been shown to harbor significantly higher CD8+ tumour-infiltrating lymphocytes (TILs), higher CD8 +/CD4 + ratios and higher PD1+ TILs compared to microsatellite-stable (MSS) OCCCs [22], and although MMR deficiency is only seen in around 10% of OCCC, some degree of tumoural and stromal PD-L1 expression has been reported in around 74% of OCCC [23], suggesting that additional subsets of OCCC show immunogenicity [20]. Notably, in the KEYNOTE−100 ovarian cancer trial, there was a trend towards improved response rates of OCCC to the checkpoint inhibitor pembrolizumab [20]. Indeed, a subgroup of OCCC with a relatively higher rate of gene mutations in the SWI/SNF complex have been shown to show an enrichment of immune-related pathway activity and poorer prognosis [24]. *ARID1A* deficiency is related to a mismatch repair-deficient phenotype with *ARID1A* mutant tumours showing an increase in TILs, activation of the immune checkpoint and sensitization to the PD-L1 checkpoint blockade in ovarian cancer in in vivo mouse models compared to *ARID1A* wild-type tumours [18]. This mechanism is thought to occur via MSH2 recruitment to the chromatin by *ARID1A* during DNA replication and mismatch DNA repair [18]. Whilst pan-histological clinical studies have highlighted a significant treatment benefit for patients with *ARID1A*-deficient tumours when treated with (PD-1)/PD-L1 immunotherapy [25], their response rate specifically in OCCC-specific trials, such as PEACOCC (NCT03425565) testing pembrolizumab in patients with advanced gynaecological clear cell cancer, has not been reported to date. Furthermore, the characterisation of immune cell types and spatial distribution within OCCC patients has not been comprehensively explored. 

Here, we sought to comprehensively characterise the immune repertoire of OCCCs at the RNA and protein level alongside histological assessment of TILs within the immune microenvironment using multiplex immunofluorescence on tissue sections in order to identify subgroups of OCCC that may potentially benefit from immune checkpoint blockade.

## 2. Materials and Methods

### 2.1. Clinical Samples

A retrospective series of primary untreated OCCC tumours (n = 34) and OCCC-like primary tumours (EAE *n* = 4; EAO *n* = 8) were obtained with appropriate ethical approval under the Royal Marsden Hospital (RMH) NHS Foundation Trust sponsored study (ID: CCR3705): “Analysis of tumour specimens for biomarkers in gynaecological cancers” (Table 1 and Appendix A). All patients provided written consent for the use of material for research purposes. Appropriate representative formalin-fixed paraffin-wax embedded (FFPE) tissue blocks were chosen based upon their histology reports. Haematoxylin and eosin (H&E) sections of each case were reviewed by two independent consultant histopathologists in order to confirm the percentage of tumour content. For each case, if a germline blood sample was unavailable, non-malignant FFPE blocks were selected and sections cut for DNA extraction. Microsatellite status was assessed by immunohistochemistry of mismatch repair (MMR) proteins MSH2, MSH6, PMS2 and MLH1 during routine clinical testing, or by PCR using fluorescent PCR-based microsatellite loci (MSI Analysis System, Version 1.2, Promega, Madison, WI, USA).

### 2.2. DNA Extraction and Library Preparation

Genomic DNA was extracted from FFPE tissue sections using the QIAamp FFPE Tissue Kit (Qiagen, Manchester, UK) based upon the manufacturer’s protocols for both tumour and non-malignant content. The extraction of genomic DNA from blood samples was completed using the QIAamp Blood mini kit (manual) or QIAsymphony DNA Midi Kit (automated) (Qiagen) using the manufacturer’s protocols. The quality of the extracted DNA was analysed using the Agilent 2200 Tapestation (Agilent, Stockport, UK) and the Qubit Fluorometer (Fisher Scientific, Loughborough, UK). Both DNA extraction and next generation sequencing (NGS) were completed at Good Clinical Laboratory Practice (GCLP)-accredited laboratories.

### 2.3. Targeted Sequencing

In order to identify common mutations in OCCC, a 59-gene targeted capture panel (originally established to target 59 genes for the FOrMAT clinical trial (Feasibility of Molecular Characterization Approach to Treatment, CCR3994, Royal Marsden NHS Hospital, Foundation Trust) (Nimblegen, Roche, Welwyn Garden City, UK)) was applied as previously described [26]. Detailed methodology regarding preparation and variant calling has been previously described [26] (Appendix A). Raw targeted sequencing data is deposited in the NCBI Sequence Read Archive under the accession PRJNA432413 and PRJNA432343. 

### 2.4. ARID1A Immunohistochemistry

Immunohistochemistry (IHC) was performed on 3–4 μm thick whole tissue sections along with H&Es for each specimen with assistance from the Breast Cancer Now Histopathology Core Facility and the Royal Marsden NHS Foundation Trust Diagnostic Laboratory. Slides were incubated with anti-ARID1A, rabbit monoclonal antibody (1:1000) and EPR13501 (Abcam, Cambridge, UK), using the Dako-Autostainer Link 48 with the EnVision FLEX kit according to the manufacturer’s instructions (Agilent Technologies, Cheadle, UK). Human breast, tonsil, appendix, prostate and kidney tissues were used as positive controls and xenograft models were obtained as previously described [19]. Stromal cells were used as an internal positive control. Cases were scored by two independent consultant pathologists for ARID1A protein expression using a modified Allred scoring system as previously described [26].

### 2.5. Histological Quantification of Immune Infiltrate

The extent of lymphocytic infiltration in H&E stained tumour sections was assessed independently by two pathologists. Both were blinded to the ARID1A sequencing and IHC findings. The Salgado scoring system, a standardised methodology originally designed for the quantification of tumour-infiltrating lymphocytes (TILs) in breast cancer, was used [27]. TILs were scored as a percentage of the stromal area alone and areas occupied by carcinoma cells were not included in the total assessed surface area [27]. The mean of the pathologists’ scores was taken to obtain an overall Salgado score for each case.

### 2.6. NanoString nCounter Profiling with the PanCancer Immune Panel

For discovery, the NanoString nCounter^®^ PanCancer Immune Profiling Panel was used to quantify mRNA abundance in primary OCCC tumours (*n* = 25), together with endometrioid carcinomas (endometrioid adenocarcinoma of the ovary (EAO) *n* = 8, and endometrioid adenocarcinoma of the endometrium (EAE) *n* = 4), given the genomic similarities of these tumour types to OCCC [28,29] and the small number of OCCC in our cohort. The panel targets 730 genes from 14 different immune cell types [30]. The input target amount of RNA was 75–80 ng, adjusted for the degree of fragmentation based upon the percentage of RNA fragments between 50–300 nucleotides of total RNA in the sample. RNA samples were hybridised to capture and reporter probes using a Thermo Cycler for 18–22 h at 65 °C. Once hybridised, probes were bound and aligned to the nCounter Cartridge and counted by the digital analyser. NanoString nCounter data were pre-processed using R package NanoStringNorm* (v1.2.1) [31], and normalised with Limma voom using R package Limma (v3.38.3) [32]. All visualisations were created using custom libraries in R statistical environment (v.3.6.0). NanoString nCounter^®^ gene expression output was normalised across all samples combined (OCCC *n* = 25, EAE *n* = 4 and EAO *n* = 8), using the edgeR package (v3.24.3) [33]. Normalisation factors were calculated using the weighted trimmed mean of M-values (TMM). Batch effects within normalised expression data were calculated and adjusted across NanoString cartridges.

### 2.7. Differential Gene Expression Analysis

Normalised gene expression counts were used to perform differential gene expression analysis. For each gene of the NanoString nCounter^®^ PanCancer Immune panel (*n* = 730), differential mRNA abundance analysis was performed using Limma in R statistical environment (v.3.6.0). Genes were considered significantly differentially expressed if satisfying |log_2_ fold change| > 1 and FDR adjusted *p*-value < 0.05. A lenient threshold of |log_2_ fold change| > 1 and *p*-value < 0.05 was also applied to identify the most statistically differentially expression genes if no genes satisfied the initial significance criteria.

Tumours from patients with recurrent disease within four years of diagnosis were assigned as high risk (*n* = 15) and tumours from patients without recurrent disease within a follow-up time of four or more years were assigned as low risk (*n* = 19) at the time of analysis (November 2020). Censored patients (i.e., samples from patients without recurrent disease and less than four years of follow-up time) were not assigned a risk category and were excluded from subsequent model training (*n* = 3). One further sample was excluded from model training due to being taken from a metastatic biopsy in a patient already included in the dataset with her primary disease.

### 2.8. Risk Predictor

Gene expression profiles of the OCCC and endometroid primary tumours profiled with the NanoString immune panel described above (Figure 1) from disease recurrence risk DGE analysis, were used to develop a supervised clustering model for prediction of risk in further datasets. Samples with a high/low risk label (*n* = 34) were used as observations to train a knn model using the R package Class (v7.3.15). mRNA abundance of training and validation datasets were scaled to z-scores for training and testing the knn. The number of neighbours included in majority voting (k) was determined by the square root of observations followed by ceiling function (k = 7). Three independent studies were used as validation datasets to validate the predictor including endometrial cancers and clear cell renal kidney cancers given reported genomic similarities between these and OCCC [28,29,34]: (1) Uehara et al. OCCC samples (*n* = 25) [35], (2) TCGA UCEC endometrioid endometrial adenocarcinoma samples (*n* = 107) [28], 3) TCGA KIRC samples (*n* = 533) [34]. Kaplan–Meier estimators were calculated using the R package Survival (v3.1.12) for each validation dataset using the knn-predicted risk labels as groups. Statistical difference between the risk groups was estimated using the log-rank test. For TCGA datasets, Disease-Specific Survival (DSS) was used for outcome analysis. For Uehara et al. [35], both Disease-Specific Survival (DSS) and Progression Free Survival (PFS) were used for outcome analysis.

### 2.9. Multiplex Immunofluorescence (MIF)

Multiplex immunofluorescence (MIF) biomarker imaging was performed on 31 OCCC samples to enable the simultaneous evaluation of six markers in a single FFPE tissue section. MIF staining was performed by sequential staining of 4 μm FFPE sections from each patient using an Opal 7−colour reagent kit (Akoya Bioscience, Marlborough, MA, USA). After de-waxing and rehydrating, the sections underwent heat-mediated antigen retrieval before staining. Antibodies were stripped after staining by repeat antigen retrieval before each new antibody was applied. The following antibodies were used: CD4 (Abcam, 133616), CD8 (Dako, M710301), PD-L1 (Cell Signaling, 13684), FOXP3 (Abcam, 20034), Pan–cytokeratin (Dako, M351501) and CD68 (Dako, M087629). Positive control tonsil tissue samples were stained for each different marker individually. Multispectral imaging was performed using the Vectra^®^ 3.0 pathology imaging system (Akoya Bioscience). Cell phenotyping and density quantification was automated using a custom algorithm developed in the inForm™ image analysis software package (Perkin Elmer, Buckinghamshire, UK).

### 2.10. Quantitative Tissue Assessment of PD-L1 Expression and Combined Positive Score Calculation Using HALO^®^ Image Analysis Platform

Assessment of PD-L1 spatial location was performed using the automated High-Plex FL module in HALO^®^ (Indica Labs) in order to calculate the combined positive score (CPS), encompassing the number of PD-L1-positive cells (tumour, lymphocytes and macrophages) in relation to total tumour cells [36].

Eighteen of the thirty cases produced a CPS from the quantification of PD-L1-positive cells. Nucleated cells were segmented using optimised nuclear detection thresholds (nuclear contrast: 0.97; nuclear intensity: 0.15; nuclear segmentation aggressiveness: 0.59; and nuclear size: 0700.885). Cells were then classified by phenotype based on cytoplasmic-positive and nuclear-positive detection thresholds. PanCK staining intensity varied across images and was grouped into 4 categories based on algorithm settings that best segmented PanCK-positive tumour cells from PanCK-negative cells. An algorithm with PD-L1 detection thresholds (nucleus: 5; cytoplasm 0.5) and PanCK detection thresholds (nucleus: 5 or nucleus: 14, and cytoplasm: 0.28/0.478/3.47/7) was applied to the images from 18 cases. 

## 3. Results

### 3.1. Immune Related Gene Expression Signatures Are Associated with Risk of Recurrence in OCCC

In order to characterise the immune landscape associations with prognosis in OCCC, we first used targeted gene expression profiling of 730 immune related genes using the NanoString nCounter^®^ PanCancer Immune Profiling Panel that were subjected to targeted DNA sequencing and immunohistochemical assessment of ARID1A protein expression (Figure 1) (OCCC *n* = 25). We additionally included OCCC-like primary tumours (*n* = 12), given the genomic similarities of these disease types to OCCC [26,27,28]. Gene expression levels were compared between (i) high-risk patients (disease recurrence within 4 years; *n* = 15), and (ii) low-risk patients (no disease recurrence with a follow-up time of at least four years; *n* = 19). Five genes passed a significance threshold of |log2 fold change| > 1, *p*-value < 0.05. The cell surface protein cluster of differentiation 24 *CD24* and the HLA class II beta chain coding gene *HLA-DRB3* were under-expressed in the high-risk group (*p* = 0.008 and *p* = 0.012, respectively) and the complement component gene *C1S,* the Fos proto-oncogene *FOS* and the adhesive glycoprotein thrombospondin-1 *THBS1* were over-expressed in the high-risk group (*p* = 0.007, *p* = 0.025 and *p* = 0.031, respectively; Figure 2A–E, Appendix A, Appendix A).

To evaluate whether these genes were prognostic, we next trained a supervised k-nearest neighbour (*knn)* clustering model (Figure 2B) for the prediction of disease-recurrence risk in independent OCCC and OCCC-like tumour cohorts. This risk predictor was tested in three independent validation datasets: (1) an independent dataset of 25 OCCC primary tumours [35]; (2) TCGA UCEC endometrioid endometrial adenocarcinoma samples (*n* = 107) [28] and (3) TCGA KIRC samples (*n* = 533) [34]. Given the lack of clinically well-annotated OCCC cohorts with survival data, gene expression data available for UCEC and KIRC were used given their biological similarities to OCCC [1,29,37]. We first predicted the risk status (high or low) for each sample in the validation datasets (Figure 2C–E). In the independent Uehara OCCC cohort and TCGA KIRC cohort, the recurrence probability was statistically significant between the risk groups (Uehara OCCC cohort: *p* = 6.2 × 10^−4^, HR = 17.03 (1.94–149.61) logrank test, TCGA KIRC cohort *p* = 0.036, HR = 1.5 (1.02–2.19) Figure 2C,E, Appendix A). Of note, estimation of immune cell subpopulations using mRNA abundance, did not identify any significant associations between the various immune subpopulations and risk status (Appendix A). 

### 3.2. ARID1A Mutated OCCC Show Differential Gene Expression of Immune Related Genes Associated with Lymphocyte Recruitment

We next compared the relative gene expression of immune related genes between *ARID1A* mutant compared to *ARID1A* wild-type tumours. We identified that *CC2L0* and *TREM1* were over-expressed in *ARID1A* mutant cases (*p* = 0.001 and *p* = 0.001, respectively), and *SERPING1* and *CXCL14* were under-expressed in this group (*p* = 0.004 and *p* = 0.02, respectively; Appendix A and Appendix A). Of note, *CCL20* is involved in lymphocyte attraction and *TREM1* and *CXCL14* are involved in attraction of neutrophils and macrophages, respectively. Comparison of immune cell subtype quantifications inferred from mRNA abundance levels, however, did not identify any significant differences in immune subpopulations according to *ARID1A* mutation status (Appendix A). 

### 3.3. OCCCs Are Characterised by Histologically Low Levels of Immune Infiltrate

We next sought to ascertain whether the level of immune cell infiltrate was associated with risk of recurrence or *ARID1A* mutation status in OCCC. We initially used gross histological assessment of immune infiltrate and distribution using the Salgado scoring system [27] in 31 primary OCCC tumours (Figure 1) and correlated TILs with FIGO tumour stage, clinical risk status and *ARID1A* mutation status (Appendix A). Overall, OCCCs were characterised by low stromal lymphocytic infiltrates, with a median percentage of TILs of 1.5% (interquartile range: 1–3%), (Figure 3A–D, Appendix A). The majority of cases, 87.1% (27/31), had stromal TIL scores of ≤ 5%, whilst 3.2% of cases (1/31) had between 5–10% TILs and 9.7% of cases (3/31) had > 10% TILs. We observed no significant association between immune infiltrate and FIGO stage, relapse and *ARID1A* mutation status (Appendix A). Additionally, we observed no associations between relapse at 4 years (risk status) and *ARID1A* mutation status (*p* = 0.3777, Fisher’s exact test, Appendix A). Of note, all cases with high immune infiltrate were microsatellite (MSS) stable.

### 3.4. Total Immune Subpopulation Cell Quantifications Are Not Associated with Risk Status and ARID1A Mutation Status in OCCC

Although OCCC tumours have overall low levels of immune infiltrate, there is emerging evidence that they may respond well to immune checkpoint blockade [20,38,39,40]. Moreover, *ARID1A* mutant tumours have been shown to demonstrate activation of the immune checkpoint pathway and sensitization to the PD-1/PD-L1 checkpoint blockade in ovarian cancer in in vivo mouse models. We thus hypothesised that in-depth quantification of various immune subpopulations with a role in the potential immune response to these agents may identify subgroups of patients associated with clinical outcomes. To investigate this, we undertook multiplex immunofluorescence using the Opal™ immunofluorescence staining system, and Vectra™ imaging systems. In total, 30 OCCCs were assessed (13 *ARID1A* mutant and 17 *ARID1A* wild-type) with antibodies against CD4 (T-helper cells and regulatory T cells), FOXP3 (regulatory T cells), CD8 (cytotoxic T cells), CD68 (monocyte and/macrophage cells), PD-L1 and PanCK (Pan Cytokeratin, which stains tumour cells) (Figure 1) and quantified using a custom algorithm developed in the inForm™ image analysis software package (see methods). Overall, we observed a significant correlation between CD8+ staining and Salgado scoring (*p* = 0.0233, Spearman’s rank test, Appendix A). Similar to the TIL quantification using Salgado methodology, we observed no association between overall levels of immune markers and risk status in this cohort. There were also no significant differences in immune markers between *ARID1A* mutant and wild-type tumours (Appendix A).

The CPS (which assesses the number of PD-L1-positive tumour, lymphocytic and macrophage cells in relation to total tumour cells) [36] has been identified as a predictive biomarker of higher overall response rate (ORR) to pembrolizumab in mixed histology epithelial ovarian cancers. Of the 18 OCCCs where CPS calculations were feasible using HALO^®^, no significant association between the CPS scores and risk status was observed (with 4/11 cases low-risk and the remaining 7/11 cases classed as high-risk, *p* = 0.4121, Mann–Whitney U test). In addition, we observed no significant associations between CPS score and *ARID1A* mutational status (*ARID1A* 7 = mutant and 11 = wild-type: *p* = 0.9503, *p* = 0.3734 *Mann–Whitney* U test, Appendix A). 

### 3.5. Spatial Distribution of Immunosuppressive Immune Subpopulations Is Associated with Risk Status and ARID1A Mutational Status in OCCC

We next sought to further interrogate the spatial locations of key immune subpopulations known to promote an immunosuppressive environment [41,42,43,44], including comparison of immune cell location amongst the tumour cells relative to the stroma. We therefore investigated the spatial locations of tumour-associated macrophages (TAMs) by assessing CD68 + cells, activated TAMs (PD–L1 + CD68 + cells), regulatory T cells (FOXP3 + CD4 + cells and activated regulatory T cells; PD–L1 + FOXP3 + CD4 + triple positive cells [45]), stratifying into tumoral and stromal locations. In low-risk patients, we identified significantly higher numbers of CD68 +, PD-L1 + CD68 + and PD-L1 + FOXP3 + cells in the stroma relative to tumour (*n* = 9) (CD68 +: *p* = 0.0117, PD-L1 + CD68 +: *p* = 0.0039, PD-L1 + FOXP3 + CD4 + *p* = 0.0039, paired samples Wilcoxon signed-rank test, Figure 4A–H). No significant differences were observed in the spatial location of these cell types in high-risk patients, however (*n* = 12) (CD68: *p* = 0.8501, PD–L1 + CD68: *p* = 0.3013, PD–L1 + FOXP3 + CD4 + *p* = 0.1055, paired samples Wilcoxon signed rank test, Figure 4A–D, Appendix A). 

Comparing the spatial locations of the immunosuppressive subpopulations with *ARID1A* mutation status highlighted that CD68 + cells were significantly more prevalent in the stroma compared to tumour in the mutant cases (*n* = 13, *p* = 0.0007), yet no significant difference between these two spatial locations was present in wild-type cases (*n* = 17, *p* = 0.1089, paired samples Wilcoxon signed rank test) (Figure 4E–H). There were no significant differences in the spatial locations of PDL1 + TAMs (PD–L1 + CD68 + cells), or regulatory T cells (Tregs) (FOXP3 + CD4 + cells and PD–L1 + FOXP3 + CD4 + cells) in *ARID1A* mutant tumours, however (Appendix A).

Taken together, these results are suggestive that the increased presence of TAMs in the stroma and subsequent exclusion from the tumour cells in low-risk and *ARID1A* mutant tumours is indicative of a reduced immunosuppressive environment in these patients.

### 3.6. Spatial Distribution of Cytotoxic T Cells Is Associated with ARID1A Mutational Status

Previous work has highlighted that *ARID1A* deficiency is related to a mismatch repair phenotype and increased TILs in ovarian cancer mouse models. Given that we did not identify higher TILs in *ARID1A* mutant cases at the histological level, and also found no differences in the total quantifications of various immune subpopulations assessed through multiplex immunofluorescence, we sought to further clarify if there was any difference in the spatial locations of cytotoxic T-lymphocyte subpopulations that mediate tumour cell killing (dual positive CD4 + CD8 + cells) between *ARID1A* mutant and wild-type cases. We found that CD8 + cells (cytotoxic T cells) were significantly more prevalent in the stroma of the *ARID1A* mutant cases relative to the tumour (*p* = 0.0034), and this coexisted with significantly higher stromal CD68 + cells (*p* = 0.0007), whilst the wild-type cases showed no significant differences between the spatial location of this cell type (*p* = 0.0984) (Figure 4E–H). There were no differences, however, between *ARID1A* mutant and wild-type cases in terms of the locations of CD4 + T cells (Appendix A). 

## 4. Discussion

By comprehensively characterising a cohort of OCCC, we have gained novel insights into the immune landscape of this histologically rare ovarian epithelial malignancy. Our study identified that although OCCCs have low levels of immune infiltrate, there is a subset of immune-related genes associated with clinical outcome in OCCC patients. Although derivation of this signature was supplemented with endometroid cancers to increase the power of the survival modelling, the signature validated in additional independent OCCC, endometrial and clear cell kidney cancers, highlighting the interoperability of the immune-related prognostic signature in cancers with reported genomic similarities [28,29,30,31,32,33,34]. Three of five genes that we identified as associated with disease recurrence (*CD24, C1S* and *THBS1)* were also found to be prognostic in a recent study by Tan and colleagues, which identified two prognostic OCCC gene expression subtypes in a large cohort of OCCCs [24], and may thus represent useful biomarkers for identification of poor prognosis OCCC tumours.

We further identify that the spatial location of immune subpopulations, rather than their gross quantification, is associated with patient risk of relapse and *ARID1A* mutation status. Our finding of relatively higher numbers of important immunosuppressive subpopulations, CD68 + and PD-L1 + CD68 + (TAMs) and PD-L1 + FOXP3 + CD4 + (T-regulatory) cells in the stroma relative to tumour in low-risk patients, suggests that the ‘tumour-exclusion’ of these cells is important in maintaining an effective anti-tumour immune response and preventing tumour progression. Recent studies have suggested that increased intratumoral PD-L1 + macrophages and T-regs are associated with a worse outcome in lung cancers [46,47]. High FOXP3 expression has additionally been associated with worse progression-free survival in high-grade serous epithelial ovarian cancer patients [48]. Whilst we did not identify this direct association in OCCC, likely due to low numbers, we provide new insights into the spatial significance of T-regulatory cells with relation to prognosis in OCCC.

When *ARID1A* mutation status was considered, we identified that mutant cases showed the simultaneous ‘tumour-exclusion’ of both CD68 + TAMs and CD8 + cytotoxic T cells. Previous studies have also demonstrated that stromal TAMs have long-lasting interactions with stromal T cells, ‘trapping’ them in the stroma and preventing them from gaining access to the tumour cells [49]. Our findings suggest that the therapeutic targeting of TAMs may be important in enabling CD8 + cells to access the tumour, further enhancing and synergising current immune checkpoint-based immunotherapy in *ARID1A* mutant patients [42,43,44,45,46,47,48,49,50]. Known synthetic-lethal targeting strategies associated with ARID1A deficiency, such as ATR inhibitors and/or PARP inhibitors [19,20,21,22,23,24,25,26,27,28,29,30,31,32,33,34,35,36,37,38,39,40,41,42,43,44,45,46,47,48,49,50,51], may also help prime the immune system and synergise the immune checkpoint blockade. Results from the ongoing ENGOT/GYNI/ATARI (ATr Inhibitor in Combination With Olaparib in Gynaecological Cancers With ARId1A Loss or no Loss (ATARI)) trial will be useful to assess this.

Our study has a number of limitations, namely, the small numbers of patient samples analysed within the cohort. Evaluation of our findings in larger cohorts and in the context of OCCC-specific clinical trials assessing the effectiveness of immunomodulatory agents (e.g., PEACOCC) will be important. In our cohort, no OCCCs harbored microsatellite instability. MSI has been reported in OCCC at a frequency of approximately 10% [22] and immunotherapy is now FDA approved for patients with MSI tumours, regardless of the primary tumour site [52]. This is thought to occur due to the high tumour mutational burden of cancers with defective mis-match repair. A recent pan-cancer analysis (not including OCCC) showed an increased mutational load in *ARID1A*-mutated tumours [53], suggesting that *ARID1A* mutant OCCCs may have a higher TMB, and thus potentially respond better to anti-PD-L1 therapies. We were unable to formally assess the TMB in this study due to the small size of the sequencing panel used, and this therefore warrants further assessment in future studies.

## 5. Conclusions

In summary, we have identified and validated an immune-related gene expression profile that is associated with a risk of recurrence that could be incorporated in future clinical trials. We have also provided new insights into the spatial significance of various immune subpopulations in OCCC, whereby tumour-associated macrophages (TAM) and regulatory T cells are excluded from the vicinity of tumour cells in low-risk patients, and TAMs and cytotoxic T cells are also excluded from the vicinity of tumour cells in *ARID1A*-mutated OCCCs. Together these findings suggest that the exclusion of these immune effectors could determine the host response of *ARID1A* mutant OCCCs to therapy. As such, *ARID1A* mutant patients may derive additional benefits from treatments targeting TAMs to further enhance the response to immunotherapy by facilitating the access of CD8 cells into the vicinity of the tumour. Together our findings provide a framework for the spatial assessment of immune subpopulations in prospective immunotherapy trials in OCCC and highlights the importance of considering combinatorial treatment approaches to improve responses to immunotherapy and overall clinical outcomes in OCCC.

## Figures and Tables

**Figure 1 cancers-13-03854-f001:**
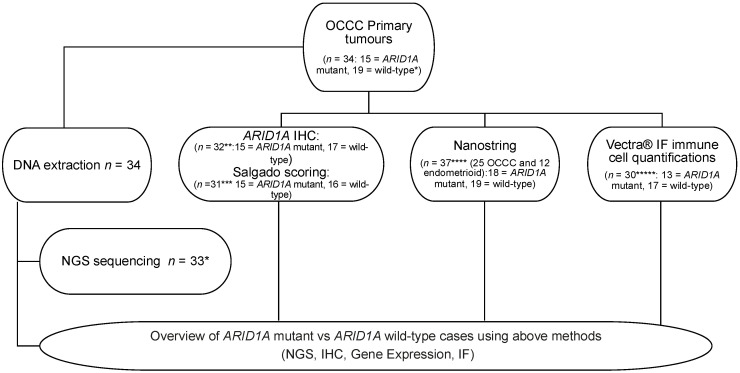
Study workflow. CONSORT diagram showing 34 ovarian clear cell carcinoma (OCCC) cases identified from the Royal Marsden biomarker study (CCR 3705). The same FFPE block was used for extracting DNA for next generation sequencing (NGS), RNA for gene expression analysis, immunohistochemistry (IHC) and multiplexed immunofluorescence (IF). * Case 3705–0573 failed NGS quality control and was not included in further analysis. ** Two of the 34 cases were not included in histological, NanoString and IF analysis: Case 3705–0573 was omitted having failed sequencing and case 3705–0611 failed IHC due to absence of tumour on tissue section and no further tissue blocks available. *** 3705–0435 had matched primary and metastatic samples and only the primary sample was analysed for TIL infiltrate. **** Three further cases had inadequate material for NanoString analysis (3705–0082, 3705–0181, 3705–0713). Three cases failed QC metrics (3705–0625, 3705–0719, 3705–0459). The cohort of 25 OCCCs was supplemented with a cohort of endometrioid carcinomas (endometrioid adenocarcinoma of the ovary (EAO) *n* = 8 and endometrioid adenocarcinoma of the endometrium (EAE) *n* = 4, total = 37 cases) for NanoString analysis. ***** Cases 3705–0719 and 3705–0464 had insufficient tissue for MIF analysis.

**Figure 2 cancers-13-03854-f002:**
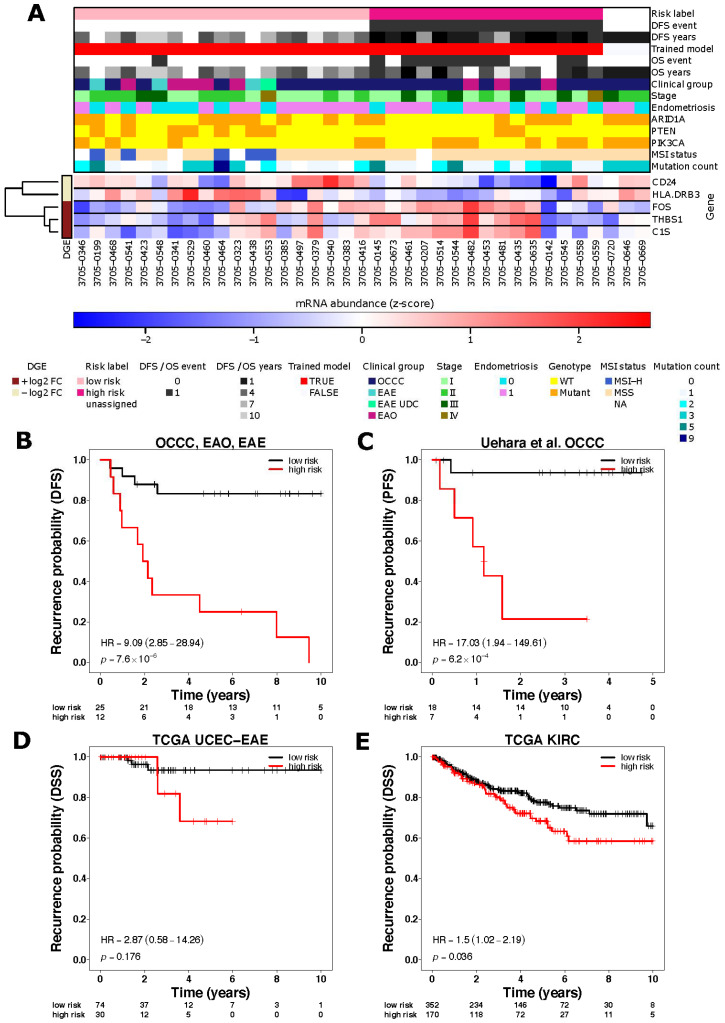
Gene Expression Profiling shows differential gene expression in low- vs. high-risk patient samples. (**A**) Heatmap demonstrating results from NanoString PanCancer Immune Panel gene expression profiling (*n* = 37, OCCC *n* = 25 and endometrioid-EAE and EAO *n* = 12). (**B**–**E**) Kaplan–Meier curves depicting the associations with prognosis of the differential gene expression signature in (**B**) our discovery cohort (all OCCC, EAE and EAO samples (*n* = 37) and independent validation cohorts), (**C**) Uehara et al. OCCC samples (*n* = 25), (**D**) TCGA UCEC endometrioid endometrial adenocarcinoma samples (*n* = 107), and (**E**) TCGA KIRC (kidney renal cell carcinoma) samples (*n* = 533). OS/DFS event 1 = is event and 0 = censored. Prognostic association (expressed as Hazard ratio ‘HR’) was estimated by fitting Cox proportional hazards model.

**Figure 3 cancers-13-03854-f003:**
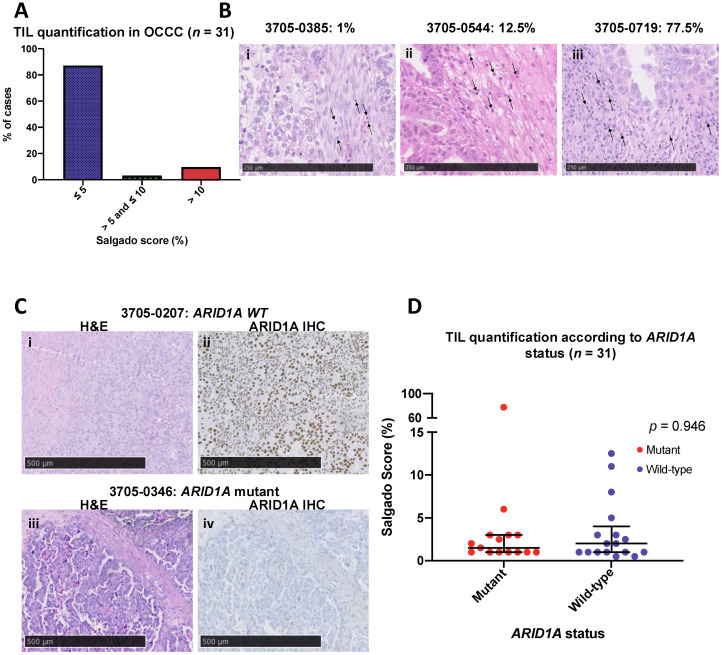
OCCC is characterized by low immune infiltrates. (**A**) Bar chart showing that the greatest proportion of cases (87.1%, 27/31) fall within the lowest Salgado score category, whilst lower proportions fall within the intermediate (3.23%, 1/31) and high Salgado score groups (9.68%, 3/31). (**B**) From left to right: (**i**) low (1%), (**ii**) intermediate (12.5%) and (**iii**) high (77.5%) Salgado scores represented in H&E sections (arrows indicate TILs, magnification = × 20). (**C**) Representative micrographs of ARID1A protein expression in OCCC tumours: (**i**) H&E and (**ii**) matched ARID1A IHC for *ARID1A* wild-type case 3705–0207, and (**iii**) H&E and (**iv**) matched ARID1A IHC for *ARID1A* mutant case 3705–0346 showing loss of *ARID1A* expression. (**D**) Scatter plot showing Salgado scores in the cohort according to *ARID1A* mutational status.

**Figure 4 cancers-13-03854-f004:**
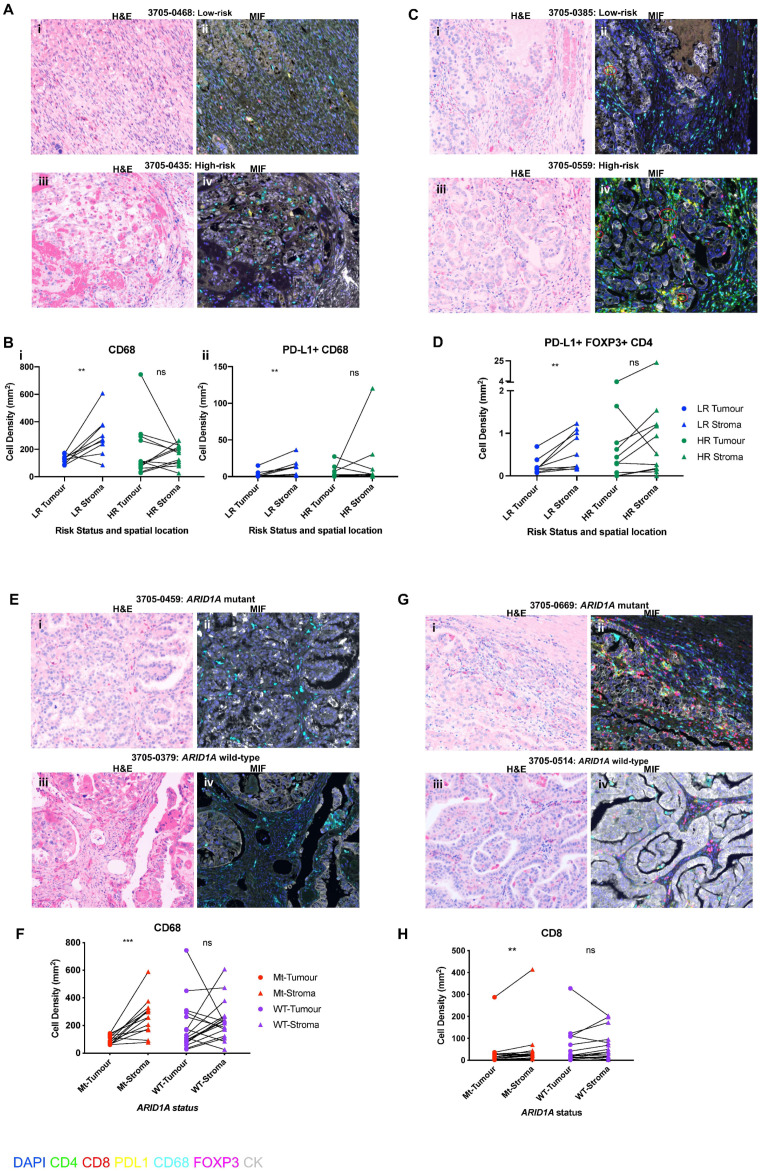
Spatial distribution of immune subpopulations is associated with risk status and *ARID1A* mutational status in OCCC. (**A**) Representative H&Es and Vectra images in low- and high-risk patients demonstrating differential spatial locations of PD-L1 + CD68 cells: (**i**) H&E and (**ii**) corresponding Vectra IF image in low-risk case 3705–0468 showing stromal PD-L1 + CD68 + cells; (**iii**) H&E and (**iv**) corresponding Vectra IF image for high-risk case 3705–0435 demonstrating PD-L1 + CD68 + cells located in a tumour area. (**B**) Immune cell quantifications subdivided into tumour and stromal locations according to patient risk status (*n* = 21). Scatter graphs depicting individual data points for each patient for cell density counts of (**i**) CD68 and (**ii**) PD-L1 + CD68. (**C**) Representative H&Es and Vectra images in low- and high-risk patients demonstrating differential spatial locations of PD-L1 + FOXP3 + CD4 + cells; (**i**) H&E and (**ii**) corresponding Vectra IF image in low-risk case 3705–0385 showing a stromal PD-L1 + FOXP3 + CD4 + cells; (**iii**) H&E and (**iv**) corresponding Vectra IF image for high-risk case 3705–0559 demonstrating PD-L1 + FOXP3 + CD4 cells located in tumour and stroma compartments. These cells are highlighted by red annotations. (**D**) Immune cell quantifications subdivided into tumour and stromal locations according to patient risk status (*n* = 21). Scatter graph depicting individual data points for each patient for cell density counts of PD-L1 + FOXP3 + CD4 + cells showing significantly higher numbers of this immune cell-type in the stroma relative to tumour in low-risk patients (*n* = 9). (**E**) Representative H&E and corresponding Vectra IF images in *ARID1A* mutant and wild-type tumours demonstrating CD68 cell findings: (**i**) H&E for mutant case 3705–0459 and (**ii**) corresponding Vectra IF image demonstrating CD68 cells located within the stroma and excluded from the tumour cells; (**iii**) H&E for wild-type case 3705–0379 and (**iv**) corresponding Vectra IF image demonstrating CD68 cells located in both the stroma and the tumour. (**F**) Scatter graph depicting individual data points depicting CD68 cell density counts in the tumour and stroma for each patient (*n* = 30). (**G**) Representative H&E and corresponding Vectra IF images in *ARID1A* mutant and wild-type cases demonstrating CD8 + cell findings: (**i**) H&E of mutant case 3705–0669 and (**ii**) corresponding Vectra IF image demonstrating CD8 + cells located within the stroma and excluded from the tumour cells; (**iii**) H&E for wild-type case 3705–0514 and (**iv**) corresponding Vectra IF image demonstrating CD8 + cells located in both the stroma and the tumour. (**H**) Scatter graph depicting individual data points for cell density counts of CD8 + cells in the tumour and stroma for each individual patient (*n* = 30), (** *p* < 0.01, *** *p* < 0.001, Wilcoxon matched pairs-signed rank test).

**Table 1 cancers-13-03854-t001:** Overview of patient characteristics in the study.

Parameter	Total Number (*n*)
**Number of primary tumour samples**	43 *
**Diagnosis**	
Ovarian Clear Cell Carcinoma (OCCC)	31
Endometrioid adenocarcinoma of the ovary (EAO)	8
Endometrioid adenocarcinoma of the endometrium (EAE)	4
**Grade**	
I	3
II	7
III	33
**FIGO Stage**	
I	17
II	14
III	10
IV	2
**Median age, years (range)**	55 (32,75)
**Endometriosis**	
Yes	24
No	15
Unknown	4

* Clinical data excludes 2 of the original 34 OCCC cases which were not analysed (3705–0573, 3705–0611 see Figure 1 legend). Patient 3705–0435 had both primary and metastatic tumour samples and clinical data relating to the primary sample are included.

## Data Availability

Raw targeted sequencing data have been deposited into the NCBI Sequence Read Archive under the accession PRJNA432413 and PRJNA432343.

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
