# Peer review of "Quantitative Assessment and Prognostic Associations of the Immune Landscape in Ovarian Clear Cell Carcinoma"

_cancers, 2021, doi:10.3390/cancers13153854_

Round 1

Reviewer 1 Report

Khalique et al. in this paper aim at comprehensively characterising the immune repertoire of OCCCs. This is an important issue, as, when OCCC is diagnosed at advanced stage, show the poorest stage-adjusted prognosis and relative resistance to chemotherapy, when compared to other EOC subtypes. Thus, there is a clear unmet clinical need to identify additional treatment for those OCCC patients that show poor responses to chemotherapy and the identification of biomarkers for OCCC patient stratification. Moreover, on one side OCCC with MMR defect are known to respond to immunotherapy with check point inhibitors. On the other side, experimental and biological evidence suggest that ARID1 gene defect (reported in 46-57% of OCCC) might be associated to MMR deficiency, although the response of OCCC and OCCC-like EOC with ARID1 mutation to checkpoint inhibitor is less clear. Unfortunately, OCCC with MMR gene defects are not included in this study and the Authors are able to evaluate a small number of OCCC carrying ARID1 defect (13/34 cases collected).

The only “positive” findings  are (i) the identification of an immune related gene expression signature associated with risk of recurrence and (ii) the discovery that ARID1A mutated OCCC show differential gene expression of immune related genes associated with lymphocyte recruitment. However, the comparison of immune cell subtype quantifications inferred from mRNA abundance levels did not identify any significant differences in immune subpopulations according to ARID1A mutation status.

Thus, most results are inconclusive and possibly negative. Indeed, the Authors report no significant association between immune infiltrate and FIGO stage, relapse and ARID1A mutation status and no associations between relapse at 4 years (risk status) and ARID1A mutation status. Results are only suggestive that the increased presence of TAMs in the stroma and subsequent exclusion from the tumours in low-risk and ARID1A mutant tumours might be  indicative of a reduced immunosuppressive environment in these patients. Another possible weakness of this work’s aim is that alternative treatment of OCCC with ARID1 defects are not available yet. However, the mentioned ATARI trial is recruiting and data could be available in the future.  

In conclusion, as the Authors themselves acknowledge, this study has a number of limitations, namely the small numbers of patient samples analyzed. However, the partial findings reported might be interesting.                             Therefore, this reviewer might only suggest to enlarge the number of cases and carry out minor corrections, as follows:

-Spell out numbers better. it has been very difficult to determine how many cases carry ARID1 mutation, and the above-mentioned numbers (13/34 OCCC) might be erroneous.

- Do not include endometrioid carcinomas in the original list: this adds misunderstanding.

- Amend some sloppiness in editing, such as parenthesis missed in the abstract and two were were in MM

Reviewer 2 Report

This is an interesting work presenting the potential clinical value of assessing the immune microenvironment in primary OC tumors to predict clinical outcomes. Some minor comments are listed below:

  • did patients with ARID1A mutated tumors perform better or worse overall in terms of DFS/OS?
  • there is somewhat conflicting evidence in the literature re: the role of ARID1a mutations and sensitivity to immunotherapy. What is the overall conclusion from this work with regards to the former statement?
  • the figures appear somewhat blurry and resolution needs to be improved.
  • The conclusions section needs to be completed with one-two statements summarizing the key findings of the study and their clinical implications.
